# Coordinate Linear Variance Reduction for Generalized Linear Programming

**Chaobing Song**[*]
University of Wisconsin-Madison
chaobing.song@wisc.edu

**Cheuk Yin Lin**[*]
University of Wisconsin-Madison
cylin@cs.wisc.edu

**Stephen J. Wright**
University of Wisconsin-Madison
swright@cs.wisc.edu

**Jelena Diakonikolas**
University of Wisconsin-Madison
jelena@cs.wisc.edu

## Abstract

We study a class of generalized linear programs (GLP) in a large-scale setting, which includes a simple, possibly nonsmooth convex regularizer and simple convex set constraints. By reformulating (GLP) as an equivalent convex-concave min-max problem, we show that the linear structure in the problem can be used to design an efficient, scalable first-order algorithm, to which we give the name *Coordinate Linear Variance Reduction* (CLVR; pronounced "clever"). CLVR yields improved complexity results for (GLP) that depend on the max row norm of the linear constraint matrix in (GLP) rather than the spectral norm. When the regularization terms and constraints are separable, CLVR admits an efficient lazy update strategy that makes its complexity bounds scale with the number of nonzero elements of the linear constraint matrix in (GLP) rather than the matrix dimensions. Further, for the special case of linear programs and by exploiting sharpness, we propose a restart scheme for CLVR to obtain empirical linear convergence. Finally, we show that Distributionally Robust Optimization (DRO) problems with ambiguity sets based on both $f$-divergence and Wasserstein metrics can be reformulated as (GLPs) by introducing sparsely connected auxiliary variables. We complement our theoretical guarantees with numerical experiments that verify our algorithm's practical effectiveness in terms of wall-clock time and number of data passes.

## 1 Introduction

We study the following generalized linear program (GLP):

$$\min_{\boldsymbol{x}} \ \left\{ \boldsymbol{c}^T \boldsymbol{x} + r(\boldsymbol{x}) : \boldsymbol{A}\boldsymbol{x} = \boldsymbol{b}, \ \boldsymbol{x} \in \mathcal{X} \right\}, \tag{GLP}$$

where $\boldsymbol{x}, \boldsymbol{c} \in \mathbb{R}^d$, $\boldsymbol{A} \in \mathbb{R}^{n \times d}$, $\boldsymbol{b} \in \mathbb{R}^n$, $r : \mathbb{R}^d \to \mathbb{R}$ is a convex regularizer, and $\mathcal{X} \subseteq \mathbb{R}^d$ is a closed convex set, such that a proximal/projection operator involving $r$ and $\mathcal{X}$ can be computed efficiently. When $\mathcal{X}$ is the nonnegative orthant $\{\boldsymbol{x} : x_i \geq 0, i \in [d]\}$ and $r \equiv 0$, (GLP) reduces to the standard form of a linear program (LP). When $\mathcal{X}$ is a convex cone and $r \equiv 0$, (GLP) reduces to a conic linear program. (GLP) is an important paradigm in traditional engineering disciplines such as transportation, energy, telecommunications, and manufacturing. In modern data science, we note the renaissance of (GLP) due to its modeling power in such areas as reinforcement learning [19], optimal transport [57], and neural network verification [39]. For traditional engineering disciplines with moderate scale or exploitable sparsity, off-the-shelf interior point methods that form and factorize matrices in each

---

[*]Equal contribution

36th Conference on Neural Information Processing Systems (NeurIPS 2022).

iteration are often good choices as practical solvers [26]. In data science applications, however, where the data are often dense or of extreme scale, the amount of computation and/or memory required by matrix factorization is prohibitive. Thus, first-order methods that avoid matrix factorizations are potentially appealing options. In this context, because the presence of the linear equality constraint in (GLP) may complicate projection operations onto the feasible set, we consider an equivalent reformulation of (GLP) as a min-max problem involving the Lagrangian:

$$\min_{\boldsymbol{x}\in\mathcal{X}\subset\mathbb{R}^d} \max_{\boldsymbol{y}\in\mathbb{R}^n} \left\{ \mathcal{L}(\boldsymbol{x},\boldsymbol{y}) := \boldsymbol{c}^T\boldsymbol{x} + r(\boldsymbol{x}) + \boldsymbol{y}^T\boldsymbol{A}\boldsymbol{x} - \boldsymbol{y}^T\boldsymbol{b} \right\}. \tag{PD-GLP}$$

In data science applications, both $n$ and $d$ can be very large. (PD-GLP) can be viewed as a structured bilinearly coupled min-max problem, where the linearity of $\mathcal{L}(\boldsymbol{x},\boldsymbol{y})$ in the dual variable vector $\boldsymbol{y}$ is vital to our algorithmic development.

## 1.1 Background

While there have been few papers that directly address (PD-GLP) — some special cases have been considered in [14, 25, 41–43, 60, 62, 63] — there has been significant recent work on first-order methods for general bilinearly coupled convex-concave min-max problems. Deterministic first-order methods include the proximal point method (PPM) [51], the extragradient/mirror-prox method (EGM) [34, 45], the primal-dual hybrid gradient (PDHG) method [15], and the alternating direction method of multipliers (ADMM) [20]. All these methods have per-iteration cost $\Theta(\mathrm{nnz}(\boldsymbol{A}))$ and convergence rate $1/k$, where $\mathrm{nnz}(\boldsymbol{A})$ denotes the number of nonzero elements of $\boldsymbol{A}$ and $k$ is the number of iterations.

For better scalability, stochastic counterparts of these methods have been proposed. [11, 33, 47, 49] have used "vanilla" stochastic gradients to replace the full gradients of their deterministic counterparts. [2, 13, 27] have exploited the finite-sum structure of the interaction term $\langle \boldsymbol{y}, \boldsymbol{A}\boldsymbol{x} \rangle$ involving both primal and dual variables to perform variance reduction. With a separability assumption for the dual variables, [3] and [16] have combined incremental coordinate approaches on the dual variables with an implicit variance reduction strategy on the primal variables. Recently, under a separability assumption for dual variables, [55] proposed a new incremental coordinate method with an initialization step that requires a single access to the full data. This approach, known as *variance reduction via primal-dual accelerated dual averaging* (VRPDA$^2$), obtains the first theoretical bounds that are better than their deterministic counterparts in the class of incremental coordinate approaches. The VRPDA$^2$ algorithm serves as the main motivation for our approach.

It is of particular interest to design algorithms that scale with the number of nonzero elements in $\boldsymbol{A}$ for at least two reasons: (i) the data matrix can be sparse; and (ii) when we consider simplified reformulations of certain complicated models, we often need to introduce sparsely connected auxiliary variables. Nevertheless, the randomized coordinate algorithms of [3, 16, 55] have $O(d)$ per-iteration cost regardless of the sparsity of $\boldsymbol{A}$. To address this issue, [24, 35] have proposed incremental primal-dual coordinate methods with per-iteration cost that scales with the number of nonzero elements in the row of $\boldsymbol{A}$ used in each iteration, at the price of needing to take a smaller step than for dense $\boldsymbol{A}$. Moreover, [5] has proposed a random extrapolation approach that admits both low per-iteration cost and larger step size. Despite these developments, all these algorithms produce less accurate iterates than the methods with $O(d)$ per-iteration cost, thus degrading their worst-case complexity. [2]

Finally, for the special case of LP, based on the positive Hoffman constant [30], [10] proved that the primal-dual formulation of LP exhibits a sharpness property that lower-bounds the growth of a normalized primal-dual gap from the same work. Leveraging this sharpness property, [10] proposed a restart scheme for the deterministic first-order methods discussed above to obtain linear convergence. [9] further extended this restart strategy using various heuristics to improve practical performance.

## 1.2 Motivation

We sharpen the focus from general bilinearly coupled convex-concave min-max problems to (GLP) and its primal-dual formulation (PD-GLP), because many complicated models can be reformulated as (GLP) and because this formulation possesses additional structure that can be exploited in algorithm

---

[2]Subsequent to this paper, a version of the PURE-CD algorithm of [5] that exploits sparsity in $\boldsymbol{A}$ was developed and analyzed in [6].

design. Our motivation for focusing on (GLP) is to bridge the large gap between the well-studied stochastic variance reduced first-order methods [7, 32, 54, 55] and the increasingly popular and complicated, yet highly structured large-scale problems arising in distributionally robust optimization (DRO) [21–23, 31, 38, 44, 52, 58, 61]; see also a recent survey by [50] and references therein.

For DRO problems with ambiguity sets defined by $f$-divergence [31, 37, 44], the original formulation is a nonbilinearly coupled convex-concave min-max problem. Even the well constructed reformulation in [37] does not admit unbiased stochastic gradients, leading to complicated algorithms and analysis. For DRO problems with ambiguity sets defined by Wasserstein metric [23, 29, 38, 52, 61], the original formulation is in general infinite-dimensional. (Finite-dimensional reformulations [23, 52] exist for special cases of logistic regression and smooth convex losses.) Solvers that have been proposed for DRO with Wasserstein metric are either multiple-loop deterministic ADMM [38] or are designed for general convex-concave problems [61].

By introducing auxiliary variables with sparse connections,[3] we show that DRO with ambiguity sets based on both $f$-divergence and the Wasserstein metric can be reformulated as (GLP). Thus, complicated DRO problems can be addressed by a simple, efficient, and scalable algorithm for (GLP). Our algorithm for solving (GLP) and the proposed reformulations of DRO are our main contributions.

## 1.3 Contributions

**Algorithm.** Motivated by VRPDA$^2$ [55], we propose a simple, efficient, and scalable algorithm for (PD-GLP). Our algorithm combines an incremental *coordinate* method with exploitation of the *linear* structure for the dual variables in (PD-GLP) and the implicit *variance reduction* effect in the algorithm, so we name it *coordinate linear variance reduction* (CLVR, pronounced "clever"). CLVR is inspired by VRPDA$^2$ but customized to the particular structure of (PD-GLP). In particular, by exploiting the fact that the max problem is linear and unconstrained in the dual variable vector $\boldsymbol{y} \in \mathbb{R}^n$, we find that the expensive initialization step used in VRPDA$^2$ is not needed and we can take simpler and larger steps. Further, in the structured case in which $\boldsymbol{A}$ is sparse and the convex constraint set $\mathcal{X}$ and the regularizer $r(\boldsymbol{x})$ are fully separable[4], we show that the dual averaging update in CLVR enables us to design an efficient lazy update strategy for which the per-iteration cost of CLVR scales with the number of nonzero elements of the selected row from $\boldsymbol{A}$ in each iteration, which is potentially much lower than the order-$d$ cost in VRPDA$^2$. Finally, CLVR uses extrapolation on dual variables rather than on primal variables considered in VRPDA$^2$, which significantly reduces implementation complexity of our lazy update strategy for structured variants of (PD-GLP). On the technical side, although both CLVR and VRPDA$^2$ are randomized algorithms that bound the primal-dual gap in expectation, the guarantee provided by CLVR is stronger as it allows bounding the expectation of the supremum gap as opposed to the supremum of expected gap in VRPDA$^2$.

To state our complexity results, we make the following scaling assumption.

**Assumption 1.** $L := \|\boldsymbol{A}\|$ and each row of $\boldsymbol{A}$ in (GLP) *is normalized with Euclidean norm* $R$.

Preprocessing in modern LP solvers [26] often ensures normalized rows/columns for the data matrix. Observe that $R \leq L \leq \sqrt{n}R$, the upper bound being achieved when all elements of $\boldsymbol{A}$ have identical value. Although the latter case is extreme, there exist ill-conditioned practical datasets where we can expect significant performance gains if the complexity can be reduced from $O(L)$ to $O(R)$. (We provide empirical comparison between the values of $L$ and $R$ in practical problems in Section 5.)

In Table 1, we give the overall complexity bounds (total number of arithmetic operations) and the per-iteration cost of a representative set of existing algorithms, including our CLVR algorithm, for solving a structured form of (PD-GLP) in which the set $\mathcal{X}$ and the function $r$ have separable structure: $\mathcal{X} = \mathcal{X}_1 \times \cdots \times \mathcal{X}_d$ with $\mathcal{X}_i \in \mathbb{R} (i \in [d])$ and $r(\boldsymbol{x}) := \sum_{i=1}^{d} r(x^i)$. To make the complexity results comparable, we assume further that for the stochastic algorithms [2, 16, 55] and our CLVR algorithm, we draw one row of $\boldsymbol{A}$ per iteration uniformly at random. The general convex setting corresponds to $r(\boldsymbol{x})$ being general convex ($\sigma = 0$), while the strongly convex setting corresponds to $r(\boldsymbol{x})$ being $\sigma$-strongly convex ($\sigma > 0$).

---

[3]"Sparse connections" here means that even though the newly introduced variables may substantially increase the problem dimensions, the number of nonzero entries in the constraint matrix remains of the same order.

[4]We state the results here for the fully separable setting for convenience of comparison; however, our results are also applicable to the block separable setting.

Table 1: Overall complexity and per-iteration cost for solving structured (PD-GLP). ("—" indicates that the corresponding result does not exist or is unknown.)

| Algorithm | General Convex (Primal-Dual Gap) | Strongly Convex (Distance to Solution) | Per-Iteration Cost |
|---|---|---|---|
| PDHG CP(2011) | $O\left(\frac{\mathrm{nnz}(\boldsymbol{A})L}{\epsilon}\right)$ | $O\left(\frac{(\mathrm{nnz}(\boldsymbol{A})+n+d)L}{\sigma\sqrt{\epsilon}}\right)$ | $O(\mathrm{nnz}(\boldsymbol{A}))$ |
| SPDHG CERS(2018) | $O\left(\frac{ndL}{\epsilon}\right)$ | $O\left(\frac{ndL}{\sigma\sqrt{\epsilon}}\right)$ | $O(d)$ |
| EVR AM (2022) | $O\left(\mathrm{nnz}(\boldsymbol{A})+\frac{\sqrt{\mathrm{nnz}(\boldsymbol{A})(n+d)nR}}{\epsilon}\right)$ | — | $O(n+d)$ |
| VRPDA$^2$ SWD(2021) | $O(nd\log\min\{\frac{1}{\epsilon},n\}+\frac{ndR}{\epsilon})$ | $O(nd\log\min\{\frac{1}{\epsilon},n\}+\frac{ndR}{\sigma\sqrt{\epsilon}})$ | $O(d)$ |
| CLVR (**This Paper**) | $O\left(\frac{\mathrm{nnz}(\boldsymbol{A})R}{\epsilon}\right)$ | $O\left(\frac{\mathrm{nnz}(\boldsymbol{A})R}{\sigma\sqrt{\epsilon}}\right)$ | $O(\mathrm{nnz}(\mathrm{row}(\boldsymbol{A})))$ |

As shown in Table 1, all the algorithms have optimal dependence on $\epsilon$ [48], while the dependence on the ambient dimensions $n, d$, the number of nonzero elements of $\boldsymbol{A}$ ($\mathrm{nnz}(\boldsymbol{A})$), and the constants $L$ and $R$ are quite different. For both the general convex and strongly convex settings and among coordinate-type methods, CLVR is the first algorithm that reduces the runtime dependence on the input matrix size from $nd$ to $\mathrm{nnz}(\boldsymbol{A})$. Moreover, the complexity of CLVR depends on the max row norm $R$ rather than the spectral norm $L$, and the per-iteration cost of CLVR depends only on the nonzero elements of the selected row from $\boldsymbol{A}$ in each iteration, which can be far less than $d$.

By exploiting the linear structure again, we provide explicit guarantees for both the objective value and the constraint satisfaction of (GLP). Further, the analysis of CLVR applies to the more general *block-coordinate* update setting, which is better suited to modern parallel computing platforms. Finally, following the restart strategy based on the *normalized duality gap* for LP introduced in [10], we propose a more straightforward strategy to restart our CLVR algorithm (as well as other iterative algorithms for (PD-GLP)): Restart the algorithm every time a widely known metric for LP optimality [8] halves. Compared with the normalized duality gap, the LPMetric can be computed more efficiently and in a more straightforward fashion.

**DRO reformulations.** When the loss function is convex, DRO problems with ambiguity sets based on $f$-divergence [44] or Wasserstein metric [23] are convex. However, because both problems either have complicated constraints or are infinite-dimensional, vanilla first-order methods are inapplicable.

For DRO with $f$-divergence, we show that by using convex conjugates and introducing auxiliary variables, the problem can be reformulated as a (GLP). As a result, the issue of biased stochastic gradients encountered in [37] does not arise, and CLVR can be applied. Even though the resulting problem has larger dimensions, due to the sparseness of the introduced auxiliary variables and the lazy update strategy of CLVR, it can be solved with complexity scaling only with the number of nonzero elements of the data matrix. Due to being cast as a (GLP), the DRO problem can be solved with $O(1/\epsilon)$ iteration complexity with CLVR, while existing methods such as [37] have $O(1/\epsilon^2)$ iteration complexity, with higher iteration cost because of the batch of samples needed to reduce bias. This improvement is enabled in part by considering the primal-dual gap (rather than the primal gap considered in [37]) and by allowing the constraints to be approximately satisfied (see Corollary 1).

For DRO with Wasserstein metric, following the reformulation of [52, Theorem 1], we show further that the problem can be cast in the form of (GLP). Compared with the existing reformulations [23, 38, 52, 61], our reformulation can handle both smooth and nonsmooth convex loss functions. In fact, our reformulation can provide a more compact form for nonsmooth piecewise-linear convex loss functions (such as hinge loss). Moreover, compared with algorithms customized to this problem [38] and extragradient methods [34, 45, 61] for general convex-concave min-max problems, our CLVR method attains the best-known iteration complexity and per-iteration cost, as shown in Table 1.

## 2 Notation and preliminaries

For any positive integer $p$, we use $[p]$ to denote $\{1, 2, \ldots, p\}$. We assume that there is a given partition of the set $[n]$ into sets $S^j$, $j \in [m]$, where $|S^j| = n^j > 0$ and $\sum_{j=1}^{m} n^j = n$. For $j \in [m]$, we use $\boldsymbol{A}^{S^j}$

to denote the submatrix of $\boldsymbol{A}$ with rows indexed by $S^j$ and $\boldsymbol{y}^{S^j}$ to denote the subvector of $\boldsymbol{y}$ indexed by $S^j$. We use $\boldsymbol{0}_d$ and $\boldsymbol{1}_d$ to denote the vectors with all ones and all zeros in $d$ dimensions, respectively. Unless otherwise specified, we use $\|\cdot\|$ to denote the Euclidean norm for vectors and the spectral norm for matrices. For a given proper convex lower semi-continuous function $f : \mathbb{R} \to \mathbb{R} \cup \{+\infty\}$, we define the convex conjugate in the standard way as $f^*(y) = \sup_{x \in \mathbb{R}}\{yx - f(x)\}$ (so that $f^{**} = f$). For a vector $\boldsymbol{u}$, the inequality $\boldsymbol{u} \geq \boldsymbol{0}$ is applied entry-wise. For a convex function $r(\boldsymbol{x})$, we use $r'(\boldsymbol{x})$ to denote an element of the subdifferential set $\partial r(\boldsymbol{x})$. The proximal operator of $r(\boldsymbol{x})$ over $\mathcal{X}$ is

$$\text{prox}_r(\hat{\boldsymbol{x}}) = \underset{\boldsymbol{x} \in \mathcal{X}}{\arg\min} \left\{ \frac{1}{2}\|\boldsymbol{x} - \hat{\boldsymbol{x}}\|^2 + r(\boldsymbol{x}) \right\}. \tag{1}$$

Further, we make the following assumptions, which apply throughout the convergence analysis.

**Assumption 2.** (PD-GLP) *attains at least one primal-dual solution* $(\boldsymbol{x}^*, \boldsymbol{y}^*)$. $\mathcal{W}^*$ *denotes the set of all primal-dual solutions.*

Due to the convex-concave property of (PD-GLP), $\mathcal{W}^*$ is a convex set in $\mathcal{X} \times \mathbb{R}^n$.

**Assumption 3.** $\hat{L} = \max_{j \in [m]} \|\boldsymbol{A}^{S^j}\|$ *is given at the input, where* $\|\boldsymbol{A}^{S^j}\| = \max_{\|\boldsymbol{x}\| \leq 1} \|\boldsymbol{A}^{S^j}\boldsymbol{x}\|$.

Note that $\hat{L}$ can be obtained either via preprocessing of the data or by parameter tuning. By combining Assumptions 1 and 3, it follows that $R \leq \hat{L} \leq \sqrt{\max_{j \in [m]}|S^j|}R$.

**Assumption 4.** $r(\boldsymbol{x})$ *is $\sigma$-strongly convex* $(\sigma \geq 0)$; *that is, for all $\boldsymbol{x}_1$ and $\boldsymbol{x}_2$ in $\mathcal{X}$ and all $r'(\boldsymbol{x}_2) \in \partial r(\boldsymbol{x}_2)$, we have $r(\boldsymbol{x}_1) \geq r(\boldsymbol{x}_2) + \langle r'(\boldsymbol{x}_2), \boldsymbol{x}_1 - \boldsymbol{x}_2 \rangle + \frac{\sigma}{2}\|\boldsymbol{x}_1 - \boldsymbol{x}_2\|^2$.*

For convex-concave min-max problems, a common metric for measuring solution quality is the primal-dual gap, which, for a feasible solution $(\boldsymbol{x}, \boldsymbol{y})$ of (PD-GLP), is defined by

$$\sup_{(\boldsymbol{u}, \boldsymbol{v}) \in \mathcal{X} \times \mathbb{R}^n} \{\mathcal{L}(\boldsymbol{x}, \boldsymbol{v}) - \mathcal{L}(\boldsymbol{u}, \boldsymbol{y})\}. \tag{2}$$

However, as the domain of $\boldsymbol{v}$ is unbounded, the primal-dual gap can be infinite, which makes it a poor metric for measuring the progress of algorithms. As a result, for measuring the progress of our algorithm, we consider the following restricted primal-dual gap instead:

$$\sup_{(\boldsymbol{u}, \boldsymbol{v}) \in \mathcal{W}} \{\mathcal{L}(\boldsymbol{x}, \boldsymbol{v}) - \mathcal{L}(\boldsymbol{u}, \boldsymbol{y})\}, \tag{3}$$

where $\mathcal{W} \subset \mathcal{X} \times \mathbb{R}^n$ is a compact (i.e., closed and bounded) convex set. The use of a restricted version of primal-dual gap is standard in the existing literature; see, e.g., [15, 46].

## 3 The CLVR algorithm

### 3.1 Algorithm and analysis for general formulation

Algorithm 1 specifies CLVR for (PD-GLP) in the general setting. The algorithm alternates the full update for $\boldsymbol{x}_k$ in Step 4 ($O(d)$ cost) with an incremental block coordinate update for $\boldsymbol{y}_k$ in Steps 5 and 6 (with $O(|S^{j_k}|d)$ cost for dense $\boldsymbol{A}$). The auxiliary variables $\boldsymbol{z}_k$ and $\boldsymbol{q}_k$ accumulate the cancellation terms in the estimation sequence and give a pathway to a straightforward development of the lazified CLVR, which appears as Algorithm 2 in the appendix. The cost of updating auxiliary vectors $\boldsymbol{z}_k$ and $\boldsymbol{q}_k$ is $O(|S^{j_k}|d)$ and $O(d)$, respectively. In essence, CLVR is a primal-dual coordinate method that uses a *dual averaging* update for $\boldsymbol{x}_k$, then updates the state variables $\{\boldsymbol{q}_k\}$ by a *linear recursion*, and computes $\boldsymbol{x}_k$ from $\boldsymbol{q}_{k-1}$ via a *proximal step* without direct dependence on $\boldsymbol{x}_{k-1}$. The output $\tilde{\boldsymbol{x}}_K$ is a convex combination of the iterates $\{\boldsymbol{x}_k\}_{k=1}^K$, as is standard for primal-dual methods. However, $\tilde{\boldsymbol{y}}_K$ is only an *affine* (not convex) combination of $\{\boldsymbol{y}_k\}_{k=0}^K$, as it involves the term $-(m-1)\boldsymbol{y}_0$ (whose coefficient is negative) and some of the coefficients $ma_k - (m-1)a_{k+1}$ multiplying $\boldsymbol{y}_k$ for $k \in \{1, \dots K-1\}$ may also be negative. An affine combination still provides valid bounds because the dual variable vector $\boldsymbol{y}$ appears linearly in (PD-GLP). Moreover, in Step 9, the term $ma_k(\boldsymbol{z}_k - \boldsymbol{z}_{k-1})$ serves to cancel certain errors from the randomization of the update w.r.t. $\boldsymbol{y}_k$, thus playing a key role in implicit variance reduction.

Theorem 1 provides the convergence results for Algorithm 1. The proof is provided in Appendix B. In the theorem (as in the algorithm), $\gamma$ is a positive parameter that can be tuned.

---

**Algorithm 1** Coordinate Linear Variance Reduction (CLVR)

---

1: **Input:** $x_0 \in \mathcal{X}, y_0 \in \mathbb{R}^n, z_0 = A^T y_0, \gamma > 0, \hat{L} > 0, \sigma \geq 0, K, m, \{S^1, S^2, \ldots, S^m\}$.
2: $a_1 = A_1 = \frac{1}{2\hat{L}m}, q_0 = a_1(z_0 + c)$.
3: **for** $k = 1, 2, \ldots, K$ **do**
4: $\quad x_k = \text{prox}_{\frac{1}{\gamma}A_k r}(x_0 - \frac{1}{\gamma}q_{k-1})$.
5: $\quad$ Pick $j_k$ uniformly at random in $[m]$.
6: $\quad y_k^{S^i} = \begin{cases} y_{k-1}^{S^i}, & i \neq j_k \\ y_{k-1}^{S^i} + \gamma m a_k(A^{S^i} x_k - b^{S^i}), & i = j_k \end{cases}$.
7: $\quad a_{k+1} = \frac{\sqrt{1+\sigma A_k/\gamma}}{2\hat{L}m}, A_{k+1} = A_k + a_{k+1}$.
8: $\quad z_k = z_{k-1} + A^{S^{j_k},T}(y_k^{S^{j_k}} - y_{k-1}^{S^{j_k}})$.
9: $\quad q_k = q_{k-1} + a_{k+1}(z_k + c) + m a_k(z_k - z_{k-1})$.
10: **end for**
11: **return** $\tilde{x}_K = \frac{1}{A_K}\sum_{k=1}^K a_k x_k, \tilde{y}_K = \frac{1}{A_K}\sum_{k=1}^K (a_k y_k + (m-1)a_k(y_k - y_{k-1}))$.

---

**Theorem 1.** *Let $x_k, y_k, k \in [K]$, be the iterates of Algorithm 1 and let $\tilde{x}_k, \tilde{y}_k$ be defined by*

$$\tilde{x}_k = \frac{1}{A_k}\sum_{i=1}^k a_i x_i, \quad \tilde{y}_k = \frac{1}{A_k}\sum_{i=1}^k (a_i y_i + (m-1)a_i(y_i - y_{i-1})), \tag{4}$$

*for $k \in [K]$. Let $\mathcal{W}_k \subset \mathcal{X} \times \mathbb{R}^n, k \in [K]$, be a sequence of compact convex sets such that $(\tilde{x}_k, \tilde{y}_k) \in \mathcal{W}_k \subset \mathcal{W} \subset \mathcal{X} \times \mathbb{R}^n$, where $\mathcal{W}$ is also convex and compact. Then:*

$$\mathbb{E}\Big[\sup_{(u,v)\in\mathcal{W}_k} \{\mathcal{L}(\tilde{x}_k, v) - \mathcal{L}(u, \tilde{y}_k)\}\Big]$$
$$\leq \frac{1}{A_k}\Big(\mathbb{E}\Big[\frac{\gamma}{2}\|\hat{u} - x_0\|^2 + \frac{1}{\gamma}\|\hat{v} - y_0\|^2\Big] + \frac{\gamma}{2}\|x^* - x_0\|^2 + \frac{1}{2\gamma}\|y^* - y_0\|^2\Big), \tag{5}$$

*where $(\hat{u}, \hat{v}) = \arg\sup_{(u,v)\in\mathcal{W}_k}\{\mathcal{L}(\tilde{x}_k, v) - \mathcal{L}(u, \tilde{y}_k)\}$. Furthermore,*

$$\mathbb{E}\Big[\frac{\gamma + \sigma A_k}{4}\|x_k - x^*\|^2 + \frac{1}{2\gamma}\|y_k - y^*\|^2\Big] \leq \frac{\gamma}{2}\|x^* - x_0\|^2 + \frac{1}{2\gamma}\|y^* - y_0\|^2. \tag{6}$$

*Define $K_0 = \lceil\frac{\sigma}{18\hat{L}m\gamma}\rceil$. Then in the bounds above:*

$$A_k \geq \max\Big\{\frac{k}{2\hat{L}m}, \frac{\sigma}{(6\hat{L}m)^2\gamma}\Big(k - K_0 + \max\Big\{3\sqrt{2\hat{L}m\gamma/\sigma}, 1\Big\}\Big)^2\Big\}.$$

Observe that $(\hat{u}, \hat{v})$ in the theorem statement exists because of compactness of $\mathcal{W}_k$ and our assumptions on $r(\cdot)$. The parameter $\gamma$ can be tuned to balance the relative weights of primal and dual initial quantities $\|x^* - x_0\|$ and $\|y^* - y_0\|$ (or estimates of these quantities), which can significantly influence practical performance of the method.

In addition to the guarantee on the variational form, due to the linear structure, we also provide explicit guarantees for both the objective and the constraints in (GLP), stated in the following corollary.

**Corollary 1.** *In Algorithm 1, for all $k \geq 1$, $\tilde{x}_k$ satisfies*

$$\mathbb{E}[\|y^*\| \cdot \|A\tilde{x}_k - b\|] \leq \frac{\gamma\|x^* - x_0\|^2 + \frac{1}{2\gamma}\|y^* - y_0\|^2 + \frac{1}{\gamma}\mathbb{E}[\|v - y_0\|^2]}{A_k},$$

$$|\mathbb{E}[(c^T\tilde{x}_k + r(\tilde{x}_k)) - (c^T x^* + r(x^*))]| \leq \frac{\gamma\|x^* - x_0\|^2 + \frac{1}{2\gamma}\|y^* - y_0\|^2 + \frac{1}{\gamma}\mathbb{E}[\|v - y_0\|^2]}{A_k},$$

*where $v = 2\frac{\|y^*\|}{\|A\tilde{x}_k - b\|}(A\tilde{x}_k - b)$.*

In CLVR, we allow for arbitrary $(x_0, y_0) \in \mathcal{X} \times \mathbb{R}^n$. Nevertheless, by setting $y = 0_n$, we can obtain $z_0 = 0_d$ at no cost — a useful strategy for large-scale problems since it avoids the (potentially expensive) single matrix-vector multiplication w.r.t. $A$.

## 3.2 Lazy update for sparse and structured (PD-GLP)

In Algorithm 1, direct computation of the iterates $(\boldsymbol{x}_k, \boldsymbol{y}_k)$ and the output points $(\tilde{\boldsymbol{x}}_k, \tilde{\boldsymbol{y}}_k)$ can be expensive. However, [18] showed that it is possible to only update the averaged vector in the coordinate block chosen for that iteration. This strategy requires us to record the most recent update for each coordinate block and update it only when it is selected again, which is tricky and needs to be implemented carefully. For sparse and block coordinate-separable instances of (PD-GLP), we show that by introducing auxiliary variables that are sparsely connected, we can significantly simplify CLVR and make its complexity scale independently of the ambient dimension $n \cdot d$, instead scaling with nnz($\boldsymbol{A}$). Due to space constraints, we defer technical details, including the lazy version of CLVR and associated proofs, to Appendix A.

## 3.3 Restart scheme

We now propose a fixed restart strategy with a fixed number of iterations per each restart epoch and discuss an adaptive restart strategy for the special case of standard-form LP, which corresponds to (GLP) with $r(\boldsymbol{x}) \equiv 0$ and $\mathcal{X} = \{\boldsymbol{x} : x_i \geq 0, i \in [d]\}$. We write

$$\min_{\boldsymbol{x}} \boldsymbol{c}^T \boldsymbol{x} \text{ s.t. } \boldsymbol{A}\boldsymbol{x} = \boldsymbol{b}, \ \boldsymbol{x} \geq \boldsymbol{0}_d, \tag{LP}$$

and the primal-dual form

$$\min_{\boldsymbol{x} \geq \boldsymbol{0}_d} \max_{\boldsymbol{y} \in \mathbb{R}^n} \left\{ \mathcal{L}(\boldsymbol{x}, \boldsymbol{y}) = \boldsymbol{c}^T \boldsymbol{x} + \boldsymbol{y}^T \boldsymbol{A}\boldsymbol{x} - \boldsymbol{y}^T \boldsymbol{b} \right\}. \tag{PD-LP}$$

This problem has a sharpness property that can be used to obtain linear convergence in first-order methods [10]. For convenience, in the following, we define $\boldsymbol{w} = (\boldsymbol{x}, \boldsymbol{y})$, $\hat{\boldsymbol{w}} = (\hat{\boldsymbol{x}}, \hat{\boldsymbol{y}})$, $\tilde{\boldsymbol{w}} = (\tilde{\boldsymbol{x}}, \tilde{\boldsymbol{y}})$ and $\boldsymbol{w}^* = (\boldsymbol{x}^*, \boldsymbol{y}^*)$. Meanwhile, for $\gamma > 0$, we denote the weighted norm $\|\boldsymbol{w}\|_{(\gamma)} := \sqrt{\gamma \|\boldsymbol{x} - \boldsymbol{x}^*\|_2^2 + \frac{1}{\gamma} \|\boldsymbol{y} - \boldsymbol{y}^*\|_2^2}$. Further, we use $\mathcal{W}^*$ to denote the optimal solution set of the LP and define the distance to $\mathcal{W}^*$ by $\text{dist}(\boldsymbol{w}, \mathcal{W}^*)_{(\gamma)} = \min_{\boldsymbol{w}^* \in \mathcal{W}^*} \|\boldsymbol{w} - \boldsymbol{w}^*\|_{(\gamma)}$. When $\gamma = 1$, $\|\cdot\|_{(\gamma)}$ is the standard Euclidean norm. Then based on (PD-LP), we can use the following classical LPMetric[5] to measure the progress of iterative algorithms for LP:

$$\text{LPMetric}(\boldsymbol{x}, \boldsymbol{y})$$
$$= \sqrt{\|\max\{-\boldsymbol{x}, \boldsymbol{0}\}\|_2^2 + \|\boldsymbol{A}\boldsymbol{x} - \boldsymbol{b}\|_2^2 + \|\max\{-\boldsymbol{A}^T\boldsymbol{y} - \boldsymbol{c}, \boldsymbol{0}\}\|_2^2 + |\max\{\boldsymbol{c}^T\boldsymbol{x} + \boldsymbol{b}^T\boldsymbol{y}, 0\}|^2}, \tag{7}$$

which can be explicitly and directly computed. For the Euclidean case ($\gamma = 1$), it is well-known [30] that there exists a Hoffman constant $H_1$ such that

$$\text{LPMetric}(\boldsymbol{w}) \geq H_1 \text{dist}(\boldsymbol{w}, \mathcal{W}^*)_{(1)}. \tag{8}$$

Using the equivalence of norms in finite dimensions, for general $\gamma > 0$, we can conclude that there exists another constant $H_\gamma$ (to which we refer as the generalized Hoffman's constant) such that

$$\text{LPMetric}(\boldsymbol{w}) \geq H_\gamma \text{dist}(\boldsymbol{w}, \mathcal{W}^*)_{(\gamma)}. \tag{9}$$

Using Eq. (9) and Theorem 1, we then obtain the following bounds for distance and LPMetric.

**Theorem 2.** *Consider the* CLVR *algorithm applied to the standard-form LP problem* (PD-LP)*, with input* $\boldsymbol{w}_0$ *and output* $\tilde{\boldsymbol{w}}_k$*. Given* $\gamma > 0$*, define* $\boldsymbol{w}^* = \arg\min_{\boldsymbol{w} \in \mathcal{W}^*} \|\boldsymbol{w}_0 - \boldsymbol{w}\|_{(\gamma)}$*, and define* $C_0 = \gamma + 1/\gamma + (\sqrt{2} + 1)\|\boldsymbol{w}_0 - \boldsymbol{w}^*\|_{(\gamma)} + \|\boldsymbol{w}^*\|_{(\gamma)}$*. Then for* $H_\gamma$ *defined as in* (9)*, we have*

$$\mathbb{E}\left[\sqrt{\text{dist}(\tilde{\boldsymbol{w}}_k, \mathcal{W}^*)_{(\gamma)}}\right] \leq 5\sqrt{\frac{\hat{L}mC_0}{H_\gamma k}}\sqrt{\text{dist}(\boldsymbol{w}_0, \mathcal{W}^*)_{(\gamma)}},$$

$$\mathbb{E}\left[\sqrt{\text{LPMetric}(\tilde{\boldsymbol{w}}_k)}\right] \leq 5\sqrt{\frac{\hat{L}mC_0}{H_\gamma k}}\sqrt{\text{LPMetric}(\boldsymbol{w}_0)}.$$

---

[5]In (PD-LP), we dualize the constraint $\boldsymbol{A}\boldsymbol{x} = \boldsymbol{b}$ by $\boldsymbol{y}^T(\boldsymbol{A}\boldsymbol{x} - \boldsymbol{b})$ instead of $\boldsymbol{y}^T(\boldsymbol{b} - \boldsymbol{A}\boldsymbol{x})$, so in our LPMetric, there exist a sign difference for $\boldsymbol{y}$ from the more common representation such as the one in [10].

As a result, by Theorem 2, if we know the values of $\hat{L}$, $\|\boldsymbol{w}^*\|_{(\gamma)}$ and $H_\gamma$, then by setting $k = \frac{100\hat{L}mC_0}{H_\gamma k}$, we can halve the square root of the distance and the LPMetric in expectation. Thus we can obtain linear convergence if we restart the CLVR algorithm after a fixed number of iterations. However, the values of $\|\boldsymbol{w}^*\|_{(\gamma)}$ and $H_\gamma$ are often unknown and thus make this strategy unrealistic in practice.

Compared with the above fixed restart strategy, a natural strategy is to restart whenever the LPMetric halves (summarized in Algorithm 4 in the appendix). Since LPMetric is easy to monitor and update, implementation of this strategy is straightforward. However, bounding the number of iterations required to halve the metric (in expectation or with high probability) seems nontrivial. What can be said (based on Theorem 2 and denoting by $K$ the number of iterations on CLVR between restarts) is that $\mathbb{P}[K > \frac{50\hat{L}mC_0}{\delta^2 H_\gamma}] \leq \delta$. This follows by Markov inequality, as $\mathbb{P}[K > k] = \mathbb{P}\Big[\sqrt{\text{LPMetric}(\tilde{\boldsymbol{w}}_k)} > \sqrt{\frac{\text{LPMetric}(\boldsymbol{w}_0)}{2}}\Big] \leq 5\sqrt{2\frac{\hat{L}mC_0}{H_\gamma k}}$. We provide a comparison between the adaptive restart scheme proposed in [10] and our proposed adaptive restart scheme in Section D.1 to demonstrate its practical competitiveness. Although we use adaptive restart in our experiments, we defer its convergence analysis to future work. Finally, as an independent and parallel work to ours, [40] proposed a high probability guarantee for scheduled restart for stochastic extragradient-type methods.

## 4  Application: DRO

Consider sample vectors $\{\boldsymbol{a}_1, \boldsymbol{a}_2, \ldots, \boldsymbol{a}_n\}$ with labels $\{b_1, b_2, \ldots, b_n\}$, where $b_i \in \{1, -1\}$ $(i \in [n])$. The DRO problem with $f$-divergence based ambiguity set is

$$\min_{\boldsymbol{x} \in \mathcal{X}} \sup_{\boldsymbol{p} \in \mathcal{P}_{\rho,n}} \sum_{i=1}^{n} p_i g(b_i \boldsymbol{a}_i^T \boldsymbol{x}), \tag{10}$$

where $\mathcal{P}_{\rho,n} = \big\{\boldsymbol{p} \in \mathbb{R}^n : \sum_{i=1}^n p_i = 1, p_i \geq 0 \ (i \in [n]), D_f(\boldsymbol{p}\|\mathbf{1}/n) \leq \frac{\rho}{n}\big\}$ is the ambiguity set, $g$ is a convex loss function and $D_f$ is an $f$-divergence defined by $D_f(\boldsymbol{p}\|\boldsymbol{q}) = \sum_{i=1}^n q_i f(p_i/q_i)$ with $\boldsymbol{p}, \boldsymbol{q} \in \big\{\boldsymbol{p} \in \mathbb{R}^n : \sum_{i=1}^n p_i = 1, p_i \geq 0\big\}$ and $f$ being a convex function [44]. The formulation (10) is a nonbilinearly coupled convex-concave min-max problem with constraint set $\mathcal{P}_{\rho,n}$ for which efficient projections are not available in general. When $g$ is a nonsmooth loss (e.g., the hinge loss), many well-known methods such as the extragradient [34, 45] cannot be used even if we could project onto $\mathcal{P}_{\rho,n}$ efficiently. However, by introducing auxiliary variables, additional linear constraints, and simple convex constraints, we can make the interacting term between primal and dual variables bilinear, as shown next. (See Appendix C for a proof.)

**Theorem 3.** *Let $\mathcal{X}$ be a compact convex set. Then the DRO problem in Eq.* (10) *is equivalent to*

$$\min_{\boldsymbol{x}, \boldsymbol{u}, \boldsymbol{v}, \boldsymbol{w}, \boldsymbol{\mu}, \boldsymbol{q}, \gamma} \ \Big\{\gamma + \frac{\rho\mu_1}{n} + \frac{1}{n}\sum_{i=1}^n \mu_i f^*\Big(\frac{q_i}{\mu_i}\Big)\Big\}$$

$$\begin{aligned}
\text{s.t.} \ \ & \boldsymbol{w} + \boldsymbol{v} - \frac{\boldsymbol{q}}{n} - \gamma\mathbf{1}_n = \mathbf{0}_n, \\
& u_i = b_i \boldsymbol{a}_i^T \boldsymbol{x}, && i \in [n] \\
& \mu_1 = \mu_2 = \cdots = \mu_n, \\
& g(u_i) \leq w_i, && i \in [n] \\
& q_i \in \mu_i \, \text{dom}(f^*), && i \in [n] \\
& v_i \geq 0, \ \mu_i \geq 0, && i \in [n] \\
& \boldsymbol{x} \in \mathcal{X}.
\end{aligned}$$

In Theorem 3, the domain of the one-dimensional convex function $f^*(\cdot)$ is an interval such as $[a, b]$, so that $q_i \in \mu_i \, \text{dom}(f^*)$ denotes the inequality $\mu_i a \leq q_i \leq \mu_i b$. Since the perspective function $\mu f^*\big(\frac{q}{\mu}\big)$ is a simple convex function of two variables, we can assume that the proximal operator for this function on the domain $\{(\mu, q) : q \in \mu \, \text{dom}(f^*), \mu > 0\}$ can be computed efficiently [12]. Similarly, we can assume that the constraint $g(u) \leq w$ admits an efficiently computable projection operator. As a result, the formulation (10) can be solved by CLVR. When expressing (10) in the form of (PD-GLP), the primal and dual variable vectors have dimensions $d + 1 + 4n$ and $3n - 1$,

respectively. However, according to Table 1, provided that $\mathcal{X}$ is coordinate separable, the overall complexity of CLVR will only be $O\left(\frac{(\mathrm{nnz}(\boldsymbol{A})+n)(R+1)}{\epsilon}\right)$.

The original DRO problem with Wasserstein metric based ambiguity set is an *infinite*-dimensional *nonbilinearly* coupled convex-concave min-max problem defined by

$$\min_{\boldsymbol{w} \in \mathbb{R}^d} \sup_{\mathbb{P} \in \mathcal{P}_{\rho,\kappa}} \mathbb{E}^{\mathbb{P}}[g(b\boldsymbol{a}^T\boldsymbol{w})], \tag{11}$$

where $\boldsymbol{a} \in \mathbb{R}^d, b \in \{1, -1\}$, $\mathbb{P}$ is a distribution on $\mathbb{R}^d \times \{1, -1\}$, $g$ is a convex loss function and $\mathcal{P}_{\rho,\kappa}$ is the Wasserstein metric-based ambiguity set [52]. Our reformulation for Eq. (11) is in Appendix C.2.

## 5 Numerical experiments

We provide experimental evaluations of our algorithm for the reformulation of the DRO with Wasserstein metric based on the $\ell_1$-norm (with $\kappa = 0.1$ and $\rho = 10$) and hinge loss. For its LP formulation (see Theorem 4 in the Appendix), we compare our CLVR method with three representative methods: PDHG [15], SPDHG [16] and PURE-CD [5]. For all algorithms we use LPMetric (7) as the performance measure and use a restart strategy based on successive halving of LPMetric (Section 3.3) to obtain linear convergence. We implemented CLVR and other algorithms in Julia, optimizing all implementations to the extent possible. Full details of the experimental setup can be found in Appendix D. Our code is available at `https://github.com/ericlincc/Efficient-GLP`.

**Comparison between values of $L$ and $R$.** As described in Section 1, a major advantage of CLVR is that the complexity of CLVR depends on the max row norm $R$ instead of the spectral norm $L$, which in the worst case for ill-conditioned problems can lead to a factor of $\sqrt{n}$ improvement. In practical problems where the problem instances are highly structured (e.g., reformulated DRO problems), $R$ can be much smaller than $L$. Table 2 provides empirical evidence for this claim. In all our experiments, we normalize each rows of $\boldsymbol{A}$ to $R = 1$ as stated in Assumption 1, so the values of $L$ demonstrate the theoretical improvements for the experiments described in Section 5.

Table 2: Values of the spectral norm $L$ in the reformulated DRO problems with Wasserstein metric after each row is normalized to $R = 1$.

| Reformulated a9a $d = 130738, n = 97929$ | Reformulated gisette $d = 44002, n = 28000$ | Reformulated rcv1 $d = 269914, n = 155198$ | Reformulated news20 $d = 5500750, n = 2770370$ |
|---|---|---|---|
| 117.3 | 65.9 | 196.4 | 1041.6 |

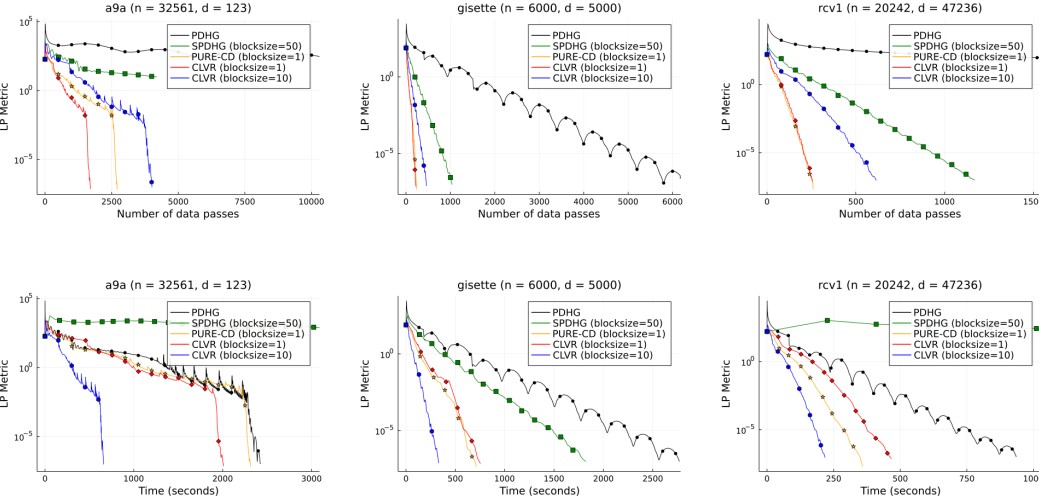

Figure 1: Comparison of numerical results in terms of number of data passes and wall-clock time.

**Comparison with primal-dual algorithms.** Figure 1 provides a comparison between algorithms in terms of the number of data passes and wall-clock time. The spikes in all the plots are due to restarts: At the beginning of each restart cycle, the value of LPMetric increases significantly, then decreases rapidly. For the number of data passes (top row), CLVR with block size 1 and PURE-CD perform best on all three datasets, CLVR with block size 10 and SPDHG with block size 50 have second-tier performance, and PDHG is worst. For the CLVR algorithm, smaller block size corresponds to smaller $\hat{L}$ in Assumption 3, which corresponds to better complexity in terms of data passes by Theorem 1. Nevertheless, the gap between empirical performance and theoretical guarantee for SPDHG and PURE-CD deserves further research because, to date, they have only been shown to have the same iteration complexity as PDHG. [6] Empirically, on a9a, CLVR with block size 1 performs better than PURE-CD in terms of data passes.

In terms of wall-clock time (bottom row of Figure 1), because of different per-iteration costs of each algorithm and instruction-level parallelism in modern processors [28], the plots differ significantly from the plots for number of data passes. Even with block size 50, SPDHG spends the most wall-clock time for one data pass and is the slowest on sparse datasets a9a and rcv1, but is faster than PDHG on the dense dataset gisette. Meanwhile, while CLVR with block size 10 is not best in terms of data passes, it remains fastest in terms of wall-clock time on all datasets due to cheaper per-iteration cost and instruction-level parallelism. On rcv1, the per-iteration cost of PURE-CD is about $60\%$ of that of CLVR with block size 1. Hence, despite having similar performance in terms of data passes, PURE-CD is faster than CLVR with block size 1, but is still slower than CLVR with block size 10.

**Comparison with production linear programming solvers.** Table 3 shows that CLVR is competitive against production-quality linear programming solvers such as GLPK [1] and Gurobi [26]. We observe that CLVR reached accurate solutions significantly faster than GLPK and Gurobi in the reformulated problems with gisette and rcv1 datasets. Although CLVR is much slower than Gurobi(barrier) on a9a dataset, we believe that much of the performance gap in this case is due to the redundancy in the problem formulation with the a9a dataset, much of which is removed by Gurobi presolver[7]. We leave presolving and other heuristic speedups of CLVR for future work.

Table 3: Comparison of numerical results between CLVR and three production solvers for linear programming, showing time required (in seconds) for each solver to reach accuracy $10^{-8}$.

| Time (seconds) | Reformulated a9a $d = 130738, n = 97929$ | Reformulated gisette $d = 44002, n = 28000$ | Reformulated rcv1 $d = 269914, n = 155198$ |
|---|---|---|---|
| JuMP+GLPK | 899 | $> 4 \times 10^4$ | $> 4 \times 10^4$ |
| JuMP+Gurobi(simplex) | 893 | 2482 | 7008 |
| JuMP+Gurobi(barrier) | **26** | 1039.7 | 1039.5 |
| CLVR | 962 | **697** | **582** |

**Conclusion.** Our preliminary numerical experiments show that CLVR is fastest in both the number of data passes and wall-clock time on considered datasets, among all primal-dual algorithms that we implemented. It is also competitive with production-quality linear programming solvers. Since it has a theoretical guarantee that matches or improves the state of the art among primal-dual methods, we believe that CLVR could be a method of choice.

## Acknowledgments and Disclosure of Funding

CS was supported in part by the NSF grant 2023239. JD and CYL acknowledge support from the NSF award 2007757. JD was also supported by the Office of Naval Research under contract number N00014-22-1-2348 and the Wisconsin Alumni Research Foundation. SW was supported by NSF grants 2023239 and 2224213, the DOE under subcontract 8F-30039 from Argonne National Laboratory, and the AFOSR under subcontract UTA20-001224 from UT-Austin. Part of this work was done while JD, CS, and SW were visiting the Simons Institute for the Theory of Computing.

---

[6] The later paper [6] describes complexity results for a newly developed version of PURE-CD that exploits sparsity in $A$.

[7] In our DRO instance with a9a dataset, Gurobi presolver removed 25% of the columns and 58% of the nonzeros.

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
