# OpenReview forum: "Coordinate Linear Variance Reduction for Generalized Linear Programming"
_NeurIPS.cc/2022/Conference — NeurIPS 2022 Accept_

### Official Review · Reviewer_FrTQ · 2022-07-09

**Rating:** 6
**Confidence:** 4
**Soundness:** 2 fair
**Presentation:** 3 good
**Contribution:** 3 good

**Summary:**

This paper develops a new primal-dual method for solving generalized linear programs (linear programs with regularizers) where each iteration requires cheap block coordinate updates. They claim an improved complexity results for linear programming and provide experiments on DRO problems.

**Questions:**

Why do you state cost per iteration of O(nnz(A) + n + d) instead of O(nnz(A)) (at many points in the paper, e.g., Line 45, Table 1)? Unless A has empty rows or columns (which can be removed from the problem) then these are equivalent. With this in mind I find the statement "CLVR is the first algorithm that has no explicit dependence on the ambient dimensions d and n, instead depending on the number of nonzero elements nnz(A)" (Line 127-128) dubious.

**Update after reading reviewer response**

Thanks for the clarification. I have a further question and comment.

How did you pick the examples? This is a very small number of examples. With such a small number and no explanation how they are picked I worry that they are cherry picked.

I think it would have been better if you'd picked a larger standard set of regression problems and presented aggregated results.

**Strengths And Weaknesses:**

I think this paper needs lots of work but has potential. In particular, if the authors can clearly demonstrate a useful generic LP solver that uses coordinate-based iterations then that would be of high value to the community. However, there are currently too many issues for me to recommend acceptance.

1. The authors present a new adaptive restart scheme. They do not give a convergence proof for this adaptive restart scheme. Also, they do not compare it empirically with existing restart schemes [9]. To be honest, I think the authors should either reuse existing adaptive restart schemes or demonstrate that their scheme is empirically better than existing schemes. I'd also remark that the results in [9] prove not only that restarts have linear convergence but the associated slope is better than what one gets is better than the linear converge rate one gets from the current iterate, and also stronger results for bilinear systems. I'd suggest the authors carefully read this paper [9] to see what I mean. I'm not sure if your restart metric would achieve these results.

2. For the numerical results please specify if PDHG uses a restart scheme and what restarts scheme it uses. You should also specify how the primal and dual step size is chosen for PDHG as this has a big impact on results e.g., see Adaptive Primal-Dual Hybrid Gradient Methods for Saddle-Point Problems by Tom Goldstein, Min Li, Xiaoming Yuan, Ernie Esser, and Richard Baraniuk.

3. I would like to have seen a much bigger variety of test problems. Currently, the method is only tested on DRO even though it is claimed to be a method for general linear programming. Ideally, you would test your method on the many standard benchmarks and report aggregate results. At a minimum you need a much larger variety of problems. Indeed, if all your method was useful for was DRO why not compare against specialized methods for DRO? E.g., Large-Scale Methods for Distributionally Robust Optimization. Daniel Levy, Yair Carmon, John Duchi & Aaron Sidford.

4. You need to cite Carmon, Yair, et al. "Coordinate methods for matrix games." 2020 IEEE 61st Annual Symposium on Foundations of Computer Science (FOCS). IEEE, 2020, and include it in Table 1. Indeed in the general convex case I believe their complexity is better.

**Minor comments**

On line 314 you write "we believe much of the performance gap in this case is due to the poor conditioning of this instance and the use of pre-conditioners in the barrier method". I don't think this correct, to my knowledge, Gurobi does not use a preconditioner in its barrier method.

On Line 257-259 you claim [38] has an adaptive scheme. It only contains a fixed frequency restart scheme.

**Update after reading reviewer response**

Thanks for the response. I think my initial score was a little harsh and appreciated your clarification. I have updated my score accordingly. I look forward to the new experiments. Please note I have updated my score irrespective of the results of the experiments (e.g., if CVR is better or not for the new instances). My only condition is the promised experiments are included in the final paper (or appendix).

---

> ### Author Response · Authors · 2022-08-02
> **Response to Reviewer FrTQ (1/2)**
>
> We thank reviewer FrTQ for their detailed comments and valuable time. We address all concerns as follows.
>
> ### General Comment.
>
> We believe that our main contribution is not in proposing a solver for “generic LP” but rather for “generalized LP” problems for which (a) standard LP solvers cannot be applied and (b) the new generation of first-order solvers is also not suitable. Hence our focus (particularly in the computational results) on the GLP problems arising from DRO, for which in both the cases of $f$-divergence and Wasserstein $1$-norm distance we have generalized LPs in which the set $\mathcal{X}$ is nontrivial - defined by convex nonlinear (possibly nonsmooth) functions. The functions defining $\mathcal{X}$ are separable, so the generalized prox-operator involving $\mathcal{X}$ is still cheap to compute, as required by our assumptions and our complexity estimates.
>
> Thus we believe that CLVR should not be compared against “generic LP” solvers but rather against other methods for saddle-point problems with bilinear coupling, which is the focus of our comparisons in the paper.
>
> We will explain these points better in a revision.
>
> ### C1. Regarding comparisons against other restart schemes.
> Let us first summarize what we show in the submitted paper. We provide a proof on the convergence rate for the fixed restart scheme if the values of $\lVert \mathbf{w}^* \rVert_{(\gamma)}$ and $H_{\gamma}$ are known (see Theorem 2 in line 244-246 and its proof in line 761-776). We argue that the assumption of such knowledge is unrealistic in practice and an adaptive restart scheme will make more sense. While we do not provide a convergence proof for the adaptive restart scheme, we prove that the number of iterations required until the LPMetric is halved can be bounded above by constants with high probability (see line 251-257). This guarantee prompts us to believe our heuristic adaptive restart scheme should achieve a linear convergence rate with high probability in practice, and we provide numerical experiments to support this claim.
>
> One main advantage of using LPMetric instead of the normalized duality gap in [9] is that LPMetric can be computed more efficiently without the overhead of solving the subproblem proposed in Section 6 of [9]. We argue that our restart scheme alleviates the issue of choosing testing frequency in practice due to evaluation costs, as there is minimal overhead caused by computing LPMetric (effectively one (sparse) matrix-vector multiplication per evaluation) even if we compute this quantity every few data passes.
>
> Unlike the normalized duality gap, the sharpness with respect to LPMetric we have shown is independent of the choice of semi-norm as described in [9], thus making our restart scheme easier to implement and more applicable to existing and future algorithms. However, we agree that further empirical comparisons against existing restart schemes would be helpful to make our claim stronger. Although we are unable to add those results in the limited time provided for the rebuttals, we can commit to adding them in the final version of the paper.
>
> ### C2. Regarding the stepsizes and restart scheme for PDHG.
> For fair comparisons, we used our new adaptive restart scheme for all the tested methods, including PDHG. For each method, we tune the step size and report the performance for the best tuned step size. Please also see lines 871-874 in Appendix D.
>
> ### C3. Regarding other test problems and comparison against existing DRO solvers.
> While reformulation of DRO as GLP is one of the main contributions of our work, we believe our method is not only useful for DRO but more generally for GLP. We are happy to include a comparison to specialized DRO solvers in a revision. We have started working on that and will update the submission if the experiments are ready by the deadline; otherwise we will include the results in the final version.
>
> ### C4. Citing Carmon, Yair, et al. "Coordinate methods for matrix games."
> Thank you for the pointer and we will include a discussion of their results. Indeed their complexity results are potentially better in some cases. But we note that this paper assumes bounded domains in $\mathbf{x}$ and $\mathbf{y}$, whereas we have unbounded domains in $\mathbf{y}$ and potentially in $\mathbf{x}$ as well.
>
> ### C5. Regarding preconditioners in Gurobi.
> Thank you for pointing out this mistake. We meant to say “presolver,” not preconditioner. We found that Gurobi’s presolver was able to reduce the problem size greatly, sometimes by factors of 2 or 3. It is possible that GLP will admit some of the same presolving techniques as LP, but this is beyond the scope of our paper.
>
> ### C6. Regarding the restart scheme in [38].
> Thank you for pointing out this mistake, which we will fix.

---

> > ### Comment · Reviewer_FrTQ · 2022-08-08
> > **General Comment**
> >
> > I understand the difference between generalized linear programming and generic linear programming (my word choice was perhaps poor) but I would rather have a generic (or general purpose) linear programming solver. I think the addition of regularizers is a pretty minor difference that usually can be handled by LP anyway (that was true for all your examples, or did I miss something?). Nonetheless, I am okay with the authors testing on a specialized set of problems (i.e., DRO) if they test against specialized baselines.

---

> > > ### Author Response · Authors · 2022-08-08
> > > **A quick clarification**
> > >
> > > This is a fair point, but let us clarify.
> > >
> > > The reason we provided experiments for problems that can be reformulated to standard LP was to also illustrate the performance with the restart strategy. So it was a nice example where we could showcase all aspects of our theoretical results while still being able to reasonably fit the whole story, including theoretical results and experiments, into 9 pages.
> > >
> > > Without restart, CLVR is applicable to broader classes of problems, but with restart we only have theoretical results for LP. Note that not all problem formulations will boil down to LP; it suffices just to take a different loss function in the DRO problems (i.e., one that is not piecewise-linear like hinge loss used in our experiments).

---

> > ### Comment · Reviewer_FrTQ · 2022-08-08
> > **C1. Regarding comparisons against other restart schemes**
> >
> > > LPMetric can be computed more efficiently without the overhead of solving the subproblem proposed in Section 6 of [9]. We argue that our restart scheme alleviates the issue of choosing testing frequency in practice due to evaluation costs, as there is minimal overhead caused by computing LPMetric (effectively one (sparse) matrix-vector multiplication per evaluation) even if we compute this quantity every few data passes.
> >
> > Given that one sparse matrix-vector product (with A and A') has been computed then the cost of computing the normalized duality gap is O(n+d) as described in Section 6.3 of [9]. Therefore given a matrix-vector product is O(nnz(A)) in practice I think there is probably a pretty minimal difference in computational time between them. At least given an optimized computation. That being said I will admit that this is simpler to implement (as you point out on line 255).
> >
> > I still stand by the fact I think this paper would have been stronger if either the authors had stuck with the original approach to doing restarts for primal-dual first-order methods, or provided an empirical comparison between these two approaches, even if this method is slightly slower than the normalized duality gap approach.

---

> > > ### Author Response · Authors · 2022-08-08
> > > **Regarding questions related to experiments**
> > >
> > > Let us first say that we are truly grateful for your constructive feedback. While it is difficult to incorporate all the suggestions in the limited time provided for the rebuttals, we do recognize that these suggestions will improve our paper and are actively working on incorporating them.
> > >
> > > Thank you for the question regarding the choice of the datasets (in the updated review). Let us clarify how we made this choice.
> > >
> > > To illustrate the performance of CLVR *with restart* (which applies to standard form LP), we needed to pick DRO problems which would boil down to this LP form (with our reformulation). An obvious choice was to use hinge loss (linear SVM with soft margin) and $\ell_1$ Wasserstein metric. Given that we end up with an SVM problem, the most obvious approach to choosing datasets was to consider LibSVM, which is a library specifically developed for SVM problems, and where datasets are all in a standardized format. The specific choice of datasets from LibSVM was then made by looking at datasets that are commonly used in ML papers and are reasonably large. a9a is a popular choice for problems that explore sparsity, because it is a sparse dataset. Gissette has been frequently used in optimization papers as it was part of the NIPS 2003 challenge. rcv1 is another frequently used dataset (see the highly cited paper “RCV1: A New Benchmark Collection for Text Categorization Research”).
> > >
> > > This all said, we are very grateful for your suggestions regarding the experiments. We are happy to report that we are in the process of running new experiments comparing the restart schemes based on the LPMetric and based on the normalized gap on the standard benchmarks also used in the Applegate et al. paper (qap10, qap15, nug08-3rd, and nug20). We hope to have some preliminary results before the rebuttal deadline, but it is hard to be certain at the moment. We will update here if we have those results on time.
> > >
> > > Regarding the experiments for DRO: this requires some more time to make sure we are making apple-to-apple comparisons. Our current implementation of CLVR is for the Wasserstein metric-based ambiguity set, while e.g., the results of Levy et al. are for CVaR and chi^2 divergence sets. We are confident we can add such experiments before the camera-ready deadline, but unfortunately the time given for rebuttals is too short to produce these results before the rebuttal deadline.

---

> > > > ### Comment · Reviewer_FrTQ · 2022-08-08
> > > > **Great!**
> > > >
> > > > Thank you for the response. I will update my score accordingly.

---

> > > > > ### Author Response · Authors · 2022-08-09
> > > > > **Thank you!**
> > > > >
> > > > > We really appreciate it.

---

> > > ### Author Response · Authors · 2022-08-09
> > > **Update on the experiments related to restart schemes**
> > >
> > > Dear Reviewer,
> > >
> > > We just wanted to share a quick update on the experiments comparing the two adaptive restart schemes (based on our LPMetric and based on the normalized gap from Applegate et al.), as we are quickly approaching the deadline. We are trying to replicate the plots from Fig. 2 in their paper, with the added lines corresponding to our restart scheme.
> > >
> > > We compared the performance of PDHG on benchmark data sets qap10, qap15, nug08, and nug20 used by Applegate et al., using the two adaptive restart criteria. We ran the algorithm until reaching accuracy as described in the Applegate et al. paper (normalized duality gap <= 1e-6 and primal & dual infeasibility <= 1e-8; extracted from their code). These are the results in terms of the iterations:
> > >
> > > ```
> > >   Dataset              |  qap10   |   qap15   |   nug08   |    nug20   |
> > >
> > >   normalized gap       |  13041   |   12561   |     841   |    22001   |
> > >
> > >   LP Metric            |  14521   |    8961   |    1481   |    16281   |
> > > ```
> > >
> > >
> > > As can be seen from the table, the two restart criteria give similar performance in terms of iteration complexity: there is no clear winner, normalized gap is better on qap10 and nug08, LP Metric is better on qap15 and nug20. A bottom line is that our restart criterion does not seem to hurt the performance in terms of iteration complexity.
> > >
> > > We are in the process of generating the plots and updating the paper comparing the two criteria over iterations. The updated paper will be available on OpenReview shortly. Note that we have not had enough time to make the plots more aesthetically appealing -- this will be done in the final version.
> > >
> > > We have not yet gotten the chance to compare in terms of time, as we have not had enough time to optimize the code for both schemes to have a fair apple-to-apple comparison. However, this is something we plan to include in the final version.

---

> > > > ### Comment · Reviewer_FrTQ · 2022-08-09
> > > > **Great!**
> > > >
> > > > Great!

---

> > ### Comment · Reviewer_FrTQ · 2022-08-08
> > **C2. Regarding the stepsizes and restart scheme for PDHG**
> >
> > > For fair comparisons, we used our new adaptive restart scheme for all the tested methods, including PDHG.
> >
> > Thanks for clarifying. Please make sure this is clear in the body of the paper.
> >
> > >  For each method, we tune the step size and report the performance for the best tuned step size. Please also see lines 871-874 in Appendix D.
> >
> > Thanks for pointing me to lines 871-874 in Appendix D. I think you need to rewrite this line so it's clear that the tuning of the weight parameter occurs for all methods. This important sentence should also be moved to the body of the paper.
> >
> > Also, please note the step size is a bit ambiguous, I think you mean to say the weight parameter in this comment or maybe primal and dual step size**s**.

---

> > > ### Author Response · Authors · 2022-08-08
> > > **StepsizeS for PDHG**
> > >
> > > We are moving some of the content about experiments from the appendix to the main body, and in particular the information about restarts and optimizing parameter choices to the best of our ability (with a pointer to full details in the Appendix). This will cause some of the text to go beyond the 9 pages, but we will deal with this post-hoc, as it requires more time to make sure other important points don't get lost in the editing process.
> > >
> > > The reason we referred to the step sizes in PDHG as the step size is that the primal and dual step sizes are fully determined by our parameter $\gamma$ and the spectral norm of the constraint matrix $A$ once we translate between PDHG and our setup (briefly, Theorem 1(b) in Chambolle-Pock tells us that the best we can do when choosing step sizes $\sigma$ and $\tau$ is to balance out primal and dual squared diameter terms while ensuring $\sigma\tau ||A||^2 < 1$). We will provide pseudocode for PDHG, SPDHG, and PURE-CD to the appendix to make sure there is no ambiguity.

---

> > ### Comment · Reviewer_FrTQ · 2022-08-08
> > **C3. Regarding other test problems and comparison against existing DRO solvers.**
> >
> > I look forward to these results. These would significantly improve the current paper.

---

> > ### Comment · Reviewer_FrTQ · 2022-08-08
> > **Citing Carmon, Yair, et al. "Coordinate methods for matrix games."**
> >
> > Thanks!
> >
> > I think you also missed:
> >
> > Alacaoglu, Ahmet, and Yura Malitsky. "Stochastic variance reduction for variational inequality methods." Conference on Learning Theory. PMLR, 2022.

---

> > > ### Author Response · Authors · 2022-08-08
> > > **Comparison to Alacaoglu-Malitsky**
> > >
> > > Thank you for pointing out this paper too. We can add a comparison to our paper.
> > >
> > > When we compare their result for the bilinear setting (page 3 in their arXiv version), we can see that this result is strictly worse than what we obtain, for the following reason. The result in Alacaoglu-Malitsky gives complexity $O(\mathrm{nnz}(A) + \frac{\sqrt{\mathrm{nnz}(A)(n + d)}M}{\epsilon})$, where $M$ is the Frobenius norm of $A$.
> > >
> > > Since we take the row norms of A to be equal to R, we have $M = ||A||_{\mathrm{Frob}} = R \sqrt{n}.$ Plugging into the above bound, the leading term is $O(\frac{\sqrt{\mathrm{nnz}(A)(n + d)n}R}{\epsilon}))$. Given that $nd \geq \mathrm{nnz}(A)$ (and, additionally, we often have $n >> d$), this bound is strictly worse than $O(\frac{\mathrm{nnz}(A)R}{\epsilon})$ obtained in our work.
> > >
> > > Of course, this paper has other results for settings that we do not consider in our work and they may also be worth mentioning.

---

> ### Author Response · Authors · 2022-08-02
> **Response to Reviewer FrTQ (2/2)**
>
> ### Q1. Regarding the per-iteration cost of PDHG.
> You are correct that the per-iteration cost of PDHG and related methods is $O(\mathrm{nnz}(\mathbf{A}))$. We will correct this estimate. With this correction, the difference between the complexities of PDHG and CLVR reduces to the difference between factors $L$ and $R$, with CLVR still holding an advantage due to the possibly smaller size of $R$.

---

> > ### Comment · Reviewer_FrTQ · 2022-08-08
> > **Thanks for the clarification!**
> >
> > Thanks for the clarification!

---

### Official Review · Reviewer_bGMS · 2022-07-09

**Rating:** 6
**Confidence:** 4
**Soundness:** 2 fair
**Presentation:** 3 good
**Contribution:** 3 good

**Summary:**

The authors of this paper proposed a scalable first-order algorithm, which is named as Coordinate Linear Variance Reduction (CLVR) for  generalized linear program (GLP). And they improved complexity results for GLP that depend on the max row norm of the linear constraint matrix in GLP rather than the spectral norm.


**Questions:**

See the weakness.

**Limitations:**

See the weakness.

**Strengths And Weaknesses:**

Strengths: This paper is well-written. It illustrates the basic idea of the CLVR method for solving GLP in a clear way. The proposed CLVR algorithm is novel. The authors establish a better complexity bound for GLP. The authors provide nice explanations and discussions for their algorithm design and theoretical analysis.

Weakness: For GLP, these papers proposed primal-dual type methods. Some comparisons are necessary.

[1] http://www.optimization-online.org/DB_FILE/2015/05/4901.pdf
[2] https://epubs.siam.org/doi/abs/10.1137/18M1229869
[3] https://arxiv.org/abs/2008.12946
[4] https://link.springer.com/article/10.1007/s40305-021-00344-x

---

> ### Author Response · Authors · 2022-08-02
> **Response to Reviewer bGMS**
>
> We thank reviewer bGMS for their great comments and valuable time. We address all concerns as follows.
>
> ### C1. Regarding the suggested citations for primal-dual methods.
> We will add comparisons with the suggested papers. A brief summary in terms of where our work improves upon the papers mentioned by the reviewer is essentially the same as when comparing to the papers summarized in Table 1 in our submission: Even though the convergence rates may be the same, dependencies on the problem-dependent quantities differ. As far as we could tell, the papers listed by the reviewer have runtimes that scale with the maximum singular value (spectral norm, or $2$-norm) of matrix $\mathbf{A}$ (called $L$ in our paper), whereas CLVR scales with the row norm $R$, which can be lower by a factor $\sqrt{n}$ on ill-conditioned instances often encountered in practice.

---

### Official Review · Reviewer_ogmY · 2022-07-11

**Rating:** 5
**Confidence:** 3
**Soundness:** 2 fair
**Presentation:** 2 fair
**Contribution:** 2 fair

**Summary:**

This work studies the generalized linear programming, which has an additional convex regularizer in the objective, in the primal-dual setting. The algorithm named CLVR was proposed to solve the problem, and its complexity depends on the number of nonzero elements in the linear constraint matrix, which makes the algorithm scalable for large-scale problems. The convergence of the algorithm was proved in the paper as well. In the special case where the problem reduces to the standard LP, a restart scheme that accelerates the algorithm was also proposed. The simulation result shows that the algorithm shows comparable and often better performance than the previous works.

**Questions:**

* As I pointed out in the weakness, the authors should advertise the originality of CLVR compared to VRPDA in a more careful manner because the problem studied in this paper is less general than [53]. To be specific, I think that the authors should answer the following question: If we change the setting of VRPDA to PD-GLP and apply the lazy update, can we get an algorithm with complexity depending only on nnz(A)?
* Please add some explanation of the variables z, q in Algorithm 1 in Section 3.1.
* What was the main difficulty or key idea in the proof of Theorem 1? How is it different from the proof of Theorem 1 or Theorem2 of [53]?

**Strengths And Weaknesses:**

Strengths: The proposed algorithm is the first one that solves PD-GLP with complexity that only depends on the number of nonzero elements in the constraint and not on any other dimensions of the problem. Theoretical convergence result is also provided.

Weakness: The work is mainly motivated by VRPDA of [53]. It seems like the proposed algorithm is the special case of VRPDA to the case where the Lagrangian is linear with respect to the dual variable, y. Thus the algorithm itself lacks originality. However, I could not check whether the analysis is also a simplification of [53] to the special case.

---

> ### Author Response · Authors · 2022-08-02
> **Response to Reviewer ogmY**
>
> We thank reviewer ogmY for their comments and valuable time. We address all concerns as follows:
>
> ### C1. Novelty issue.
> CLVR is not simply a special case of VRPDA2 [53], instead it adapts [53] to the GLP setting, simplifying and modifying the algorithm and obtaining complexity improvements over [53] (see Table 1). The complexity results are not immediate and require separate analysis, not just a simplification of the VRPDA2 analysis to the case of y appearing linearly without constraints in the PD formulation. Lines 94-110 explain the novelty of CLVR compared to VRPDA2 in detail. The main points we make here are (i) the expensive initialization step (requiring multiplications by the full matrix $\mathbf{A}$ and its transpose) are not required; (ii) we can take larger and simpler steps; (iii) we can implement CLVR is a way that takes full advantage of sparsity in $\mathbf{A}$ (while doing the same for VRPDA2 is either not possible or requires a much more complicated implementation); (iv) CLVR uses extrapolation on dual variables rather than on primal variables considered in VRPDA2, which significantly reduces implementation complexity of our lazy update strategy for structured variants of PD-GLP. Moreover, the approximate optimality guarantee provided by CLVR is stronger as it bounds the *expectation of the supremum gap* as opposed to the supremum of expected gap in VRPDA2. (See further discussion on this last point below.)
>
> ### Q1. Lazifying VRPDA2.
> The answer is that it might be possible to get a “lazy” version of VRPDA2 that adapts to sparsity in the special case of PD-GLP, but it would likely be much more complicated than what we designed for CLVR. Indeed, our initial attempts at “lazification” yielded much more complicated strategies than the one described in this paper. We found however that by introducing a single auxiliary vector (“$\mathbf{r}$” in Algorithm 2), the lazy strategy could be described and implemented quite neatly, as well as achieving the desired dependence on $\mathrm{nnz}(\mathbf{A})$. Because of the nonlinearities introduced by the presence of the function $h^*(\mathbf{y})$ in the general setting of VRPDA2, it is unlikely that a similar strategy could be designed for that case.
>
> The lazification strategy is particularly effective for solving the novel reformulated GLP instances of DRO problems as those GLP instances are highly structured and sparse (see Table 3 in Appendix D).
>
> ### Q2. Explanation for variables $\mathbf{z}$ and $\mathbf{q}$ in Algorithm 1
> The auxiliary variables $\mathbf{z}$ and $\mathbf{q}$ accumulate the cancellation terms in the estimation sequence and give a pathway to a straightforward development of the lazified CLVR, which appears as Algorithm 2 in the appendix. We will add an explanation to Section 3.1 concerning the role of these variables.
>
> ### Q3. Regarding the key idea of Theorem 1
> CLVR uses extrapolation on dual variables (see line 9 of Algorithm 3) rather than on primal variables in VRPDA2 (see line 7 of Algorithm 1 in [53]). This key change allows us to greatly simplify the analysis in our main convergence result - Theorem 1 - over the corresponding result in [53]. Moreover, importantly, it allows us to obtain a guarantee on the *expectation of the supremum gap* rather than the supremum of the expected gap, as in [53]. To see why the former bound is stronger,  consider the following toy example: A sequence of random variables that takes values $\pm 1$ with equal probability has supremum of expectation zero, but expectation of the supremum 1. Significant technical work is required to handle additional error terms - see Lemmas 2 and 5 and the proof of Theorem 1.
>
> We hope that we addressed your concerns about our contributions and that you would consider increasing your score. Please let us know if there is any additional information that we could share and that you would find useful.

---

### Official Review · Reviewer_aEEZ · 2022-07-11

**Rating:** 6
**Confidence:** 4
**Soundness:** 3 good
**Presentation:** 3 good
**Contribution:** 3 good

**Summary:**

There are many highly structured large-scale problems that can be reformulated as generalized linear programs (GLP). Hence, this work focuses on the GLP in a large-scale setting. Inspired by previous work, they proposed an up-gradation version after reformulating (GLP) as an equivalent convex-concave min-max problem, claiming lower complexity and stronger guarantee. Meanwhile, the author shows that modeling framework distributionally robust optimization with ambiguity sets defined by f-divergence and Wasserstein metric can be reformulated to GLP.

**Questions:**

I put my question in previous section.

**Limitations:**

No potential negative societal impact.

**Strengths And Weaknesses:**

Strengths：
This is a good extension of the previous work (VRPDA).
As the author claimed, better per-iteration complexity and bounding.
The solid improvement on complexity with respect to nnz(A). It’s potentially a huge improvement for certain applications.

Weaknesses：
In one of the experiments that CLVR got defeated by the Gurobi, the author claimed that they believe is due to the preconditioners. It could be better if they could provide further experiments and analyses to make the claim stronger.

---

> ### Author Response · Authors · 2022-08-02
> **Response to Reviewer aEEZ**
>
> We thank reviewer aEEZ for their positive comments and valuable time.
>
> ### Q1: Regarding “preconditioners” (actually “presolvers”) in Gurobi.
> There is a typo here as pointed out by reviewer FrTQ – We were referring to the presolvers used in Gurobi (not preconditioners). We observed that the presolvers in Gurobi could reduce the problem size by up to a factor of 3 and the number of nonzeros by a factor of 2. For the reformulated DRO instance with dataset a9a, Gurobi’s presolver removed 25% of the columns and 58% of the nonzeros.  We will conduct further experiments to support this claim. The referee’s question raises the issue of whether it is possible to use presolving in our GLP formulation. Provided that the set $\mathcal{X}$ and regularizer $r(\mathbf{x})$ had appropriate structure, many techniques used in LP presolving could be adapted for GLP. However, presolving involves an extensive array of techniques and an investigation into this issue is beyond the scope of the current paper.

---

> > ### Comment · Reviewer_aEEZ · 2022-08-06
> > **Response to the author**
> >
> > I thank the author for the response.
> > I agree with you that 'many techniques used in LP presolving could be adapted for GLP, and that takes time beyond discussion session.'
> > For the experiment that you claimed, 'We will conduct further experiments to support this claim.' I'm looking forward to it if that's possible before the discussion ends.
> >
> > However, in the other thread of comments, the per-iteration cost of PDHG and related methods have a correction. As you claimed, 'the difference between the complexities of PDHG and CLVR reduces to the difference between factors L and R. Although CLVR still holds advantages since $R \leq L \leq \sqrt{n} R$. It could be better if you manage to provide further detail to measure 'how much is this advantages.'
> >
> > Meanwhile, I believe the other reviewers did point out several important issues, and I'm also looking forward to it. (Hopefully before discussion ends)

---

> > > ### Author Response · Authors · 2022-08-06
> > > **Response about the improvements over full gradient methods**
> > >
> > > Thank you very much for asking this question and giving us the chance to clarify further!
> > >
> > > Our motivation for initially stating the per-iteration cost of PDHG as $\mathrm{nnz}(A) + n + d$ was to highlight the dense vector updates which cause the runtime to scale with $n \times d$ (instead of $\mathrm{nnz}(A)$) when directly using one of its coordinate variants such as SPDHG. However, we acknowledge that this point was lost in the process of preparing the paper and we fully agree that it is better to state the bound as $\mathrm{nnz}(A)$ and discuss this issue separately. We note, however, that among algorithms from the same class (i.e., primal-dual algorithms that use coordinate-type updates and apply to GLP), CLVR is the first method that scales with $\mathrm{nnz}(A)$ instead of $n \times d$.
> > >
> > > In terms of improving the dependence from $L$ to $R$: as discussed in Lines 114-166 of our submission, the $\sqrt{n}$ factor is achieved in the extreme case when (almost) all elements of the matrix $A$ have the same value. Even though this is an extreme case, ill-conditioned problems where the ratio of $L$ to $R$ is large are frequently encountered in practice. This is one of the reasons we see the performance gap between PDHG and CLVR in our experiments. We note that this difference between the spectral norm and either row or column norm was also the main motivation for introducing randomized coordinate methods for smooth (primal-only) minimization in Nesterov's seminal 2012 paper ("Efficiency of Coordinate Descent Methods on Huge-Scale Optimization Problems") and is the primary reason that methods such as ACDM and APCG often outperform Nesterov's (full-vector) Fast Gradient Method in practice. Finally, we note that classical coordinate methods such as APCG are inapplicable to our case, as they would need to deal with general linear constraints using projection operators, which are computationally prohibitive unless the constraints are very structured (such as, e.g., interval constraints or simplex constraints). This is the main motivation behind using a primal-dual method for GLP.
> > >
> > > We apologize that we haven't gotten the chance to update the paper on OpenReview yet. We will upload a version that addresses most of the writing & literature review concerns shortly, with the updated text shown in blue. Unfortunately, addressing other comments such as those related to the experiments requires more time, as we were asked to implement more algorithms, some requested additional discussion requires more thought and reorganization of the paper to fit into 9 pages, while some of our authors are traveling at the moment.

---

### Author Response · Authors · 2022-08-09
**The paper is now updated with additional experimental details requested by the reviewers**

Dear Reviewers,

Thank you for your constructive feedback which has greatly helped us improve our paper.

We wanted to provide a brief update regarding some of the requested additions related to the numerical experiments (all new material provided in Appendix D "Experiment Details"):

* Based on the suggestion by Reviewer aEEZ, we have added a table (Table 3 in Section D.1) that displays the values of the spectral norm $L = ||\mathbf{A}||$ when rows are normalized to have norm $R = 1$, which corroborates our theoretical claim (improvement in complexity from $L$ to $R$) for the experiments described in Section 5 (Numerical Experiments and Discussion).

* Based on the suggestion by Reviewer FrTQ, we have added a subsection (Section D.2) that compares the adaptive restart schemes from our work (based on the LP Metric) and the work by Applegate et al. (based on the normalized gap introduced in the same work), for the standard benchmarks also used in the work by Applegate et al. The results show that the adaptive restart scheme used in our work is competitive with the adaptive restart scheme from Applegate et al. in terms of iteration complexity. Comparison in terms of wall clock time will be added in the final version of the paper, but we expect some advantage of the LP Metric-based scheme as it only requires one sparse matrix-vector multiplication (vs one sparse matrix-vector multiplication plus additional $O(n+d)$ required for computing the normalized gap). Whether there is any visible advantage or the difference is minimal between the two schemes will be confirmed once we have optimized both schemes to have a fair apple-to-apple comparison.

We thank you all once again for helpful feedback and stimulating discussions.

Authors

---

### Meta-Review · Area_Chair_so1E · 2022-08-25

**Recommendation:** Accept
**Confidence:** Less certain

**Metareview:**

Overall all reviewers were positive about this paper and I tend to agree, but no reviewer felt particularly excited about the results.

**Award:**

No

---

### Decision · Program_Chairs · 2022-09-14

Accept